# Velody 2—Resilient High-Capacity MIDI Steganography for Organ and Harpsichord Music

**Eric Järpe** [1,*] and **Mattias Weckstén** [2]

1    Department of Intelligent Systems and Digital Design, Halmstad University, 301 18 Halmstad, Sweden
2    Department of CERES, Halmstad University, 301 18 Halmstad, Sweden; mattias.wecksten@hh.se
*    Correspondence: eric.jarpe@hh.se; Tel.: +46-729-773-626

**Abstract:** A new method for musical steganography for the MIDI format is presented. The MIDI standard is a user-friendly music technology protocol that is frequently deployed by composers of different levels of ambition. There is to the author's knowledge no fully implemented and rigorously specified, publicly available method for MIDI steganography. The goal of this study, however, is to investigate how a novel MIDI steganography algorithm can be implemented by manipulation of the velocity attribute subject to restrictions of capacity and security. Many of today's MIDI steganography methods—less rigorously described in the literature—fail to be resilient to steganalysis. Traces (such as artefacts in the MIDI code which would not occur by the mere generation of MIDI music: MIDI file size inflation, radical changes in mean absolute error or peak signal-to-noise ratio of certain kinds of MIDI events or even audible effects in the stego MIDI file) that could catch the eye of a scrutinizing steganalyst are side-effects of many current methods described in the literature. This steganalysis resilience is an imperative property of the steganography method. However, by restricting the carrier MIDI files to classical organ and harpsichord pieces, the problem of velocities following the mood of the music can be avoided. The proposed method, called Velody 2, is found to be on par with or better than the cutting edge alternative methods regarding capacity and inflation while still possessing a better resilience against steganalysis. An audibility test was conducted to check that there are no signs of audible traces in the stego MIDI files.

**Keywords:** MIDI; velocity values; carrier file; stego file; capacity; steganalysis resilience; audibility; file-size change-rate; mean absolute error; peak signal-to-noise ratio

## 1. Introduction

Steganography provides means for hiding information, not just making it intelligible by encrypting it. The concealment of a message at all can be the difference between life and death in cases when the very sending of a message (encrypted or not) is considered a crime and a threat to the authorities. The techniques of steganography have sometimes been criticized for serving criminals seeking to operate outside the law, but the use of it for whistleblowers and for freedom fighters (e.g., [1]) who are in countries with authoritarian regimes is well documented.

The technique of steganography does not by itself change a message, but merely hides its existence in other information. This is what distinguishes steganography from cryptography. Nevertheless, steganography is very often used in combination with cryptography, by first encrypting a message and then hiding it. This combination makes a very strong protection against revealing a secret message since upon looking for hidden messages it may be impossible to perform cryptanalysis on all possibly hidden data found in any of a great number of files. Thus, in effect, adding an encryption step can greatly improve security aspects of the message exchange—the content of the communication is not only secret but even the very existence that any kind of communication took place is unknown. This is an additional property that can be crucial in some circumstances where

the occurrence of encrypted messages can draw attention from the authorities. Since the percentage of users of steganography is unknown to a much larger extent than is the case for e.g., cryptography, it is harder to motivate its relevance [2]. This is the reason why there are few reports on the numbers of use of such methods. However, this does not mean that methods of steganography are not used.

Steganography may be deployed in many respects, but in modern times it usually means involving computer files. This study focuses on musical steganography through the MIDI format. The MIDI format is a standard music protocol used worldwide to create music and to facilitate its accessibility.

### 1.1. Related Literature

Ever since 2000, the MIDI format has been subject to methods of steganography. Worth noting are e.g., pioneer works of Inoue and Matsumoto [3] and Adli and Nakao [4]. In the former, which was preceded by several conference papers by the same authors, three requirements for steganography of MIDI files are established. These requirements are (1) that MIDI music sounds the same after steganography as it did before, (2) that the stego MIDI file should satisfy the requirements of the MIDI format, and (3) that extraction of the hidden message from the stego MIDI file should be very difficult without the proper stego key. The authors continue to outline a method for encoding data in MIDI files using permutations of note events. In the latter, three methods of steganography are briefly specified. 2009 Yamamoto and Iwakiri [5] made a short but dense paper where they present a cunning method to implement the hidden message by LSB modifications of durations of notes. This gives a high capacity, relatively speaking, for hiding messages compared to the size of the carrier file. It is claimed that little performance quality is lost which is demonstrated with a $\chi^2$-test. An ambitious study was made in Wu and Chen [6] and just recently pursued by Liu and Wu [7] about a method which modifies the way delta-times (i.e., the time elapsed between MIDI events) may be represented in the MIDI format. This renders them high capacity methods which, in the latter paper, is also claimed to be performance preserving, which means that no distortion of any kind is added to the MIDI file upon steganography with that method. Nevertheless, it inflates the MIDI file substantially and is thus not property preserving. Another recent contribution to the field is Wu, Hsiang and Chen [8] where an extremely cautious variant of velocity modification is defined by preserving the common increasing or decreasing trends in velocity among sequences of notes with increasing and decreasing pitch. There are also many methods that deal with steganography of files of the MP3 format, e.g., [9–12]. These methods may be interesting to compare to from a property perspective. For instance capacity and audibility may be considered for any music format regardless of which technique is used. Still, many comparisons are difficult because of different formats.

Aspects of capacity, robustness and transparency are mentioned in Lang et al [13]. This is an extensive inventory of the techniques for steganography in general and it even touches upon steganalysis. A more recent survey study is given by Sumathi, Santanam and Umamaaheswari [14] which also considers various steganography methods, not just audio.

### 1.2. Aspects of Steganalysis Resilience

Picture the steganalyst trying to make out whether or not a particular MIDI file is a case of steganography or not. Then, many properties would be more revealing while others are entirely plausible in a stego MIDI file. A few examples of this are:

A.  Inflation of a MIDI file making the size grow upon steganography is an obvious problem. Hiding the secret message in non-audio related parts of the MIDI file does not leave any audible traces but is still among the simplest kinds of music steganography to detect. Examples of this are Adli and Nakao [4], Wu and Chen [6] and Liu and Wu [7].

B.  In Vaske, Weckstén and Järpe [15] there are two values of the velocity throughout the stego MIDI file. This is quite unnatural to appear in any MIDI file since the MIDI

music is mainly entered in one of two ways: either by automatically scanning notes which are translated to MIDI code by some software which would make only one value of velocity (i.e., all velocity values would be identical), or the music would be played on some MIDI keyboard and entered by some MIDI sequencer program thus resulting in many different velocity values.

C.   A few methods leave audible traces (such as clicks or chirps) in the stego file. This is less common in steganography of MIDI files but does occur in music steganography of other formats, such as in Szczypiorski and Zydecki [12] and Adli and Nakao [4].

The method proposed by Liu and Wu [7] is based on the technique of altering the coding of delta times and use this encoding for permutation-based data encoding. While the method has no performance effect it does severely inflate the file size and introduce redundant data that does not contribute to the performance, which would most likely trigger a steganalyst.

By Wu, Hsiang and Chen [8] a method based on the technique of adjusting the velocities of some of the note-on events was proposed. While the goal of this strategy is to make the adjustments in such a way that the performance effect is minimized, it still changes the velocity values and is therefore classified as having a performance effect, even if not registered by mere listening to the music.

In Vaske, Weckstén and Järpe [15] a method based on a simple technique of adjusting the velocities of note-on events up or down one bit was introduced. While impossible to register such a minuscule change for a listener it is considered to have a performance effect and will also leave a telltale pattern of suspect artefacts.

The method proposed by Wu and Chen [6] manipulates the coding of the delta-time events and code data directly into these events. This strategy inflates the file size according to the authors and introduces a minuscule performance effect within a set tolerance.

In Yamamoto and Iwakiri [5] a method which manipulates the duration between events to encode data was proposed. The authors show experimental evidence of naturally occurring fluctuations which would allow the embedding to take place without being noted as suspicious. However, the suggested strategy does introduce a minuscule performance effect.

The LSB method suggested by Adli and Nakao [4] simply encodes the clear text message in the LSB of the velocity of the note-on event. This strategy does introduce a minuscule change in the performance effect, but also since no intermediate step of processing the data exists there will be non-random patterns in the LSB of the velocity values which could be detected.

The repeated command method proposed by Adli and Nakao [4] encodes data using repeating commands configured in such a way that only the last command of a series will affect the output from the interpreted MIDI file. This will inflate the file size and show up as suspect artefacts.

In Adli and Nakao [4] a SysEx method that encodes data in non-standard commands that would normally not contribute to the interpreted MIDI file output was proposed. This strategy inflates the file and shows up as suspect artefacts. The authors also claim that although output is normally not affected, in some cases there is a notable performance effect. Since this strategy does not adhere to the MIDI file standard it would also violate the second rule of SMF steganography, requiring that "The stego SMF flawlessly satisfies the specification of the standard MIDI files" as described in [3].

The method proposed by Inoue, Suzuki and Matsumoto [3] encodes data by the permutation of the order of notes in simulnotes. This strategy does not inflate the file size nor does it have any performance effect, and the two suggested strategies of permutation tries to mimic two common standards of event arrangement in the simulnotes to avoid steganalysis.

The method purposed in this paper is to develop a balanced picture of what aspects are more important in MIDI steganography and put the suggested method for MIDI steganography, Velody 2, into its scientific context among alternative methods.

## 2. The Proposed Method

The suggested method Velody 2 consists of encoding encrypted data at high capacity into the velocities of the note-on events, while mimicking humanization available in tools with MIDI support such as Ableton Live [16]. The source code of the proposed method is publicly available at http://github.com/wecksten/Velody-2.0 as referred to in Appendix A. It achieves the properties of being blind, high capacity and provides steganalysis resilience of data embedding.

The effect from using this method is that it sets the velocities to values within a narrow interval to be specified, thus removing potential mood swings in the velocity parameter of the music. This effect is minimized by restricting the use of it for organ and harpsichord music (which naturally is performed with close to constant velocities). Therefore, suspicion from steganalysts is avoided regarding the audible change of velocities. Nevertheless, the method can be used with little artificial effect on a wider range of music, as indicated by including the piano piece Für Elise by Ludwig van Beethoven in the set of MIDI songs in the experiment to test for audibility effects of the method. Of course, it is not restricted to single-instrument music, but restricting it to organ and harpsichord merely means requiring that the steganography is performed only on the velocities of these instruments though they could be a part of a larger ensemble. An example of music for an ensemble with multiple instruments is Cantata Cantata Gott der Herr ist Sonn und Schild by J.S. Bach which was part of this study.

Regarding the property of performance preservation, using Velody 2 for steganography of organ and harpsichord music should not change performance to any extent that leads to suspicions from steganalysis. As for the property of reversibility, if the point with this is to be able to show a MIDI file which does not contain any hidden message once having extracted it, this can be achieved in other ways. Therefore, this property is regarded as less important compared to the properties of steganalysis resilience and capacity for instance.

To embed a plaintext message in a carrier MIDI file the process can be split up into three steps: (1) data preparation, (2) data encryption, and (3) data encoding. The extraction process of a plaintext message from a stego MIDI file is performed in a very similar manner by reversing the order of the steps (4) data decoding, (5) data decryption, and (6) data unboxing.

### 2.1. Preparation

To be able to extract just the embedded message and nothing more, the message length needs to be known. This can be done in many ways, but assuming that most messages will be less than 256 bytes of length one approach is to add an eight-bit message header to indicate the message length. This approach allows for longer messages if that would be required by stacking several blocks after one another. Assuming most messages are less than two blocks in length this approach will have the same or less overhead than an approach where 16 bits would be used to indicate the message length. To prepare the data for encryption the clear text message $M$ of length $w$ bytes where $w = |M|$, $M$ is divided into $n$ blocks $B_1, B_2, \ldots, B_n$, where $n = \lceil \frac{w}{256} \rceil$. An eight-bit header $H_i$ is introduced for each block $B_i$, where $H_i = |B_i|$ and where $|B_i|$ is the block size in bytes. As can be seen in the Figure 1, the prepared message $P$ is equal to the assembly $P = (S, H_1, B_1, H_2, B_2, \ldots, H_n, B_n)$.

### 2.2. Encryption

The prepared message $P$ is encrypted using a standard synchronous stream cypher and a shared key generating the encrypted message $E = F(P)$ which is very similar to random data in distribution.

### 2.3. Encoding

The carrier MIDI file is unpacked into a stream of MIDI messages $S_1, S_2, S_m$ where each message of the type "note-on" is evaluated for data embedding. If the velocity for the

"note-on" event $S_i$ is less than $2^{N_e}$ the velocity $v_i = velocity(S_i)$ is replaced with a random number between a lower bound $l$ and an upper bound $u = 2^{\lfloor \log v_i \rfloor + 1} - 1$. If the velocity for the "note-on" event $S_i$ is greater than or equal to $2^{N_e}$ the velocity is LSB encoded with the next $N_e$ bits from the encrypted message stream. LSB encoding is performed by clearing the $N_e$ least significant bits of the velocity value and then adding the $N_e$ bit long value from the encrypted message stream.

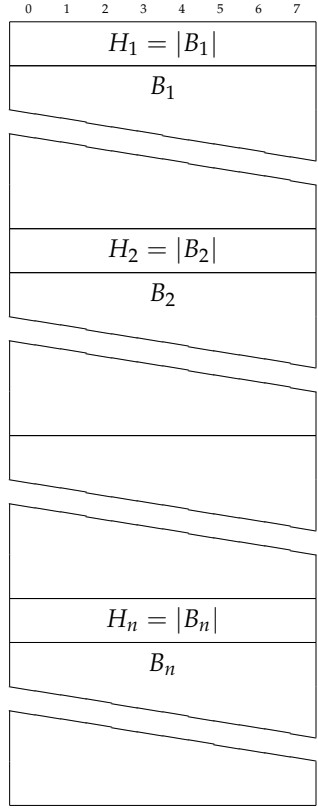

**Figure 1.** Data after preparation step. Each block of the message has been assigned a header that is equal to the following block size in bytes.

### 2.4. Decoding

The stego MIDI file is unpacked into a stream of MIDI messages $S_1, S_2, S_m$ where each message of the type "note-on" is evaluated for data decoding. If the velocity for the "note-on" event $S_i$ is greater than or equal to $2^{N_e}$ the velocity is LSB decoded using the $N_e$ least significant bits. The LSB decoding is performed by pushing the $N_e$ least significant bits of the velocity to the decoded data stream.

### 2.5. Decryption

The decoded message $D$ is decrypted using a standard synchronous stream cypher and a shared key generating the prepared message $P = F(E)$.

### 2.6. Unboxing

The prepared message $P = (P_1, P_2, \ldots, P_n)$ is unboxed by reconstruction of the header value $H_i = P_j$ and then copying $H_i$ bytes of data from the prepared message $P$ to the clear text message $M$ by adding the extracted data to the end of the clear text message $M = M + (P_{j+1}, P_{j+2}, \ldots, P_{j+H_i})$. This process is repeated until the prepared message $P$ is out of data or the header value read from the prepared message is equal to 0. The full clear text message is now available in $M$.

### 2.7. Aspects of Authenticity

In [8], the authenticity of MIDI files is paid a lot of the focus. There, the increasing or decreasing velocities due to increasing or decreasing pitch of tones played is preserved. That approach renders the method a top score with regards to authenticity.

It is assumed that the MIDI music is sequenced either by some kind of automatic procedure, where all notes are given a constant velocity throughout the whole piece or by being played by somebody, possibly at half-speed, which then results in all notes having velocities in a broad interval and relatively few velocity values being exactly the same. Therefore, the predecessor to the proposed method, Velody [15] suffers a great deal in terms of authenticity upon inspection of the actual velocity values, since these velocities are to equal proportions either of two velocity values. Thus, close inspection of the velocities by a meticulous steganalyst would reveal clear deviance from both the stereotype patterns (either all one velocity value throughout the piece or a variety of velocities).

### 2.8. Hypothesis Test of Audibility

A drawback of a steganography method is if the method leaves audible marks in the music. To this end, a hypothesis test was carried out. In this test, people listened to 10 pairs of musical pieces selected from a database of classical MIDI music (please, refer to Appendix A for a description of the database). The songs selected were

1. bach1.mid: Cantata Cantata Gott der Herr ist Sonn und Schild, part 5, BWV 79e by J.S. Bach
2. bach2.mid: Trio Sonata no 1 for organ, part 1, BWV 525a by J.S. Bach
3. bach3.mid: Trio Sonata no 1 for organ, part 2, BWV 525b by J.S. Bach
4. bach4.mid: Toccata and Fugue in D minor for organ, BWV 565 by J.S. Bach
5. bach5.mid: Prelude and Fugue in E major and C major for organ, BWV 566 by J.S. Bach
6. bach6.mid: Prelude in D major for organ, BWV 568 by J.S. Bach
7. scarlatti1.mid: Sonata Allegrissimo in C major for harpsichord, K 100 by D. Scarlatti
8. scarlatti2.mid: Sonata Allegro in A major for harpsichord, K 101 by D. Scarlatti
9. scarlatti3.mid: Sonata Allegro in G major for harpsichord, K 102 by D. Scarlatti
10. beethoven.mid: Bagatelle no 25 in A minor, "Für Elise" for piano by L. van Beethoven

The experiment was presented at a web page (see Figure 2) where all 10 pairs of musical pieces were playable via buttons embedded into the page. For each pair of songs, there was one called File 1 and another called File 2. One of these was the original song and the other was the same song but modified through Velody 2 with a message hidden inside. For each pair, the listener was instructed to guess which of the files that were steganography by indicating this using radio buttons. Each pair consisted of two MIDI songs converted to flac-format to be playable more independently on different computers. Thus, the test was solely devoted to finding out audible differences between the original song and the corresponding stego song. It did not involve the inspection of eventlists or any other kind of analysis. The number of respondents who contributed to this experiment was 30. At the top of the page, apart from declaring their name, each participant should write a 7 character code which ensured that they had been given instructions about what to do and which also served to reduce the risk of the same person taking the test multiple times and being able to differentiate between groups of respondents in retrospect.

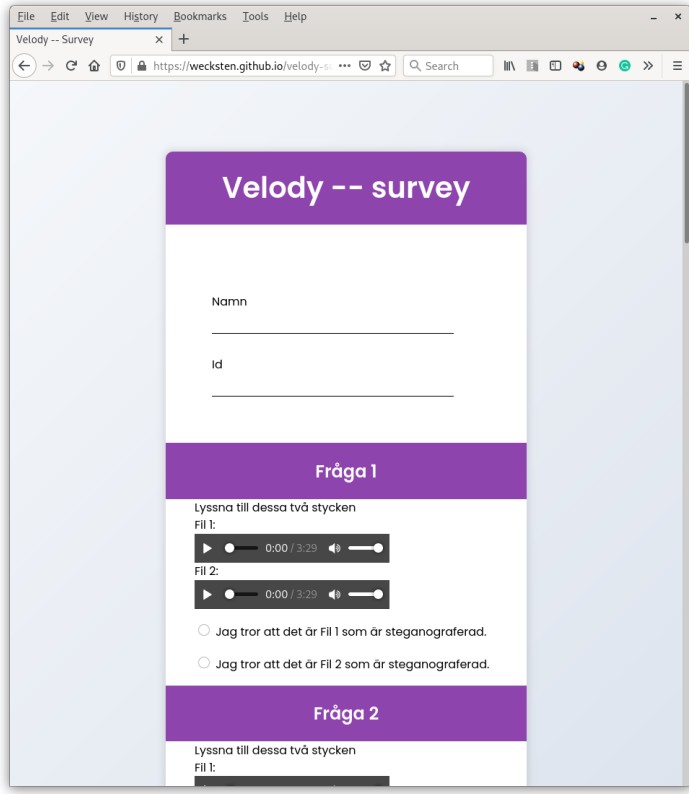

**Figure 2.** The experiment was carried out by encouraging people to listen to pairs of MIDI songs played by pressing buttons in a web form and indicating utilizing radio buttons which of the two alternatives included steganography.

### 2.9. Relevance of Power of a Hypothesis Test

If there was a detectable difference in the songs after steganography had been performed, this would be indicated by increasing the probability $\pi$ of correctly guessing which song included steganography, i.e., making this probability $> \frac{1}{2}$. If, on the other hand, there were no signs of steganography manipulation at all, that probability $\pi$ would be equal to $\frac{1}{2}$. Now, a hypothesis test could only prove the alternative, which in this case would be that $\pi > \frac{1}{2}$ as opposed to the null hypothesis $\pi = \frac{1}{2}$. The claim that there is no audible effect of Velody can never be proved by a hypothesis test. If the null is accepted this just means that effect could not be proved.

However, one might hypothesize, if there were audible effects due to the steganography method these effects would have to be so large that they resulted in a rejection of the null hypothesis. With a larger number of respondents this ability to prove an effect also if $\pi$ was just a little larger than $\frac{1}{2}$, i.e., a large number of respondents would increase the power of the test.

### 2.10. Robustness

Another aspect of importance is robustness as considered by e.g., Lang et al [13]. Currently, the proposed method, Velody 2, is not implemented with any support for improving the robustness of the method. Including the hidden message with redundancy in the carrier would not contribute to robustness since MIDI file would not play and there would not even be an eventlist in the case of as much as one bit failing. However, one could just send the stego MIDI file with redundancy, i.e., sending the same stego MIDI file multiple times. In addition, the receiver could be assumed to have many alternative addresses so the transmission could still be made to many different addresses. This way of increasing robustness is claimed to lead to minimal suspicion. Still, this is rather recommended behaviour than a part of the Velody 2 method.

### 3. Results

The results are divided into those regarding security aspects, mainly steganalysis resilience aspects such as whether there are no audible revealing footprints of the steganography and other changes which may catch the attention of an alert steganalyst, and capacity aspects, such as embedding capacity, the number of bits per event and file-size change rate, following the definitions by Liu and Wu [7].

### 3.1. Steganalysis Resilience

The Velody 2 method proposed in this paper is based on a slight extension of a velocity LSB embedding algorithm, but where the strategy to embed the data tries to mimic the output of "Humanization" available in midi tools such as Abelton. The strategy does not inflate the file size and has a minuscule performance effect. However, it has been shown in a statistical experiment that the performance effect introduced is most likely not possible to detect for a human listener. A summary of these methods and their resilience in these respects is summarized in the Table 1.

**Table 1.** Summary of properties regarding resilience to steganalysis and to what extent these are satisfied by the different methods considered.

| Method | Inflation | Performance Effect | Other Suspect Artifacts |
|---|---|---|---|
| Liu and Wu [7] | x | - | x |
| Wu, Hsiang and Chen [8] | - | x | - |
| Vaske, Weckstén and Järpe [15] | - | x | x |
| Wu and Chen [6] | x | x | - |
| Yamamoto and Iwakiri [5] | - | x | - |
| Adli and Nakao:LSB [4] | - | x | x |
| Adli and Nakao:Repeated command [4] | x | - | x |
| Adli and Nakao:SysEx [4] | x | x | x |
| Inoue, Suzuki and Matsumoto [3] | - | - | - |
| Velody 2 (the proposed method) | - | x | - |

From this table, the method in Inoue, Suzuki and Matsumoto [3] comes out best since it has no performance effect at all, it leaves a minimal amount of artificial traces while still not contributing to inflation. Other good methods are Liu and Wu [7], Wu, Hsiang and Chen [8], Yamamoto and Iwakiri [5] and Velody 2, the proposed method which are considered to suffer from only one of the shortcomings. Depending on which of these properties are more important these methods could be differently preferable.

In attempting to make a steganography method resilient it is important to leave as few traces of manipulation of the carrier upon performing the data hiding according to the method.

For instance drastically changing properties such as Mean Absolute Error (MAE), Peak Signal-to-Noise Ratio (PSNR) or file-size change-rate ($F_r$) in the information hiding process are shortcomings in that method in respect of steganalysis resilience. In Table 2 some values of these entities for a few methods are given. Here, it turns out that Velody 2 (the proposed method) and the method in Inoue, Suzuki and Matsumoto [3] are preferable with respect to file-size change-rate. Values of MAE and PSNR could not be compared since no such values were found for the other methods in the literature.

**Table 2.** Table of Mean Absolute Error (MAE), Peak Signal-to-Noise Ratio (PSNR) and file-size change-rates ($F_r$) for the methods considered. The file-size change-rates are as defined by Liu and Wu [7] and briefly explained in the text. Values within parentheses are standard errors.

| Method | MAE | PSNR | $F_r$ |
|---|---|---|---|
| Velody 2, 4 bits | 6.27 (0.36) | 24.64 (0.43) | 0.00% (0.00%) |
| Velody 2, 5 bits | 12.52 (0.81) | 18.71 (0.50) | 0.00% (0.00%) |
| Velody 2, 6 bits | 25.37 (1.88) | 12.62 (0.55) | 0.00% (0.00%) |
| Wu, Hsiang and Chen [8] | unknown | 24.99 [2] (0.60) | small |
| Liu and Wu [7] | unknown | unknown | 40.47% [1] |
| Inoue, Suzuki and Matsumoto [3] | unknown | unknown | 0.00% (0.00%) |

[1] According to Liu and Wu [7]. [2] This is WSNR which is different to PSNR but still a variant of a Signal-to-Noise Ratio.

### 3.2. Audibility

There were 30 respondents to the audibility form which consisted of telling which of two alternatives of the same song was steganography for 10 different MIDI songs. The 10 songs were as listed in Section 2.8 and the results of this experiment are illustrated in the bar charts in Figure 3.

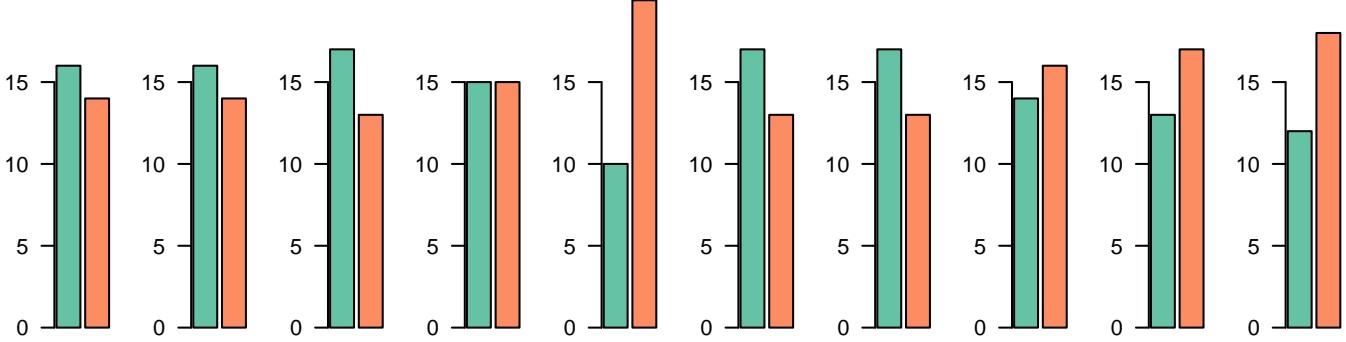

**Figure 3.** Barplots of the distribution of guesses divided into the 10 songs in the audibility experiment. A total of 30 respondents guessed each and every of the 10 pairs of songs about which one was steganography. The bars show the number of incorrect guesses in turquoise and the number correct guesses in orange for each pair of songs.

We tested the hypothesis that the suggested method left audible traces a test of whether the probability $\pi = P$ (a respondent cannot tell apart the stego MIDI file from the carrier MIDI file) exceeds 0.5 against the null hypothesis that the probability $\pi = 0.5$ (corresponding to the respondent choosing one of the alternatives entirely at random).

Letting each guess be coded as 0 if it is wrong and 1 if it is right, a binomial test is an obvious possibility since the sum $S$ of correct guesses over all pairs of MIDI songs and all respondents is a sum of 0-1-variables which, assuming independence between songs and respondents and that all guesses are correct with equal probability $\pi$, is binomially distributed with parameters $N$ and $\pi$. 30 respondents were signing up for the experiment and 10 pairs of songs making $n = 300$. In total, 153 correct guesses made the *p*-value of the binomial test

$$P(S > 153 \mid H_0) = \left. \sum_{k=154}^{N} \binom{N}{k} \pi^k (1-\pi)^{N-k} \right|_{\substack{N=300 \\ \pi=0.5}} = 0.3431$$

which is well and truly above than any reasonable level of significance, i.e., there are no indications of standing a better chance of guessing which of the songs is steganography after listening to both MIDI carrier and stego MIDI file.

The more common test for this kind of question is a $\chi^2$-test. Letting each song constitute a class, the number of correct guesses for each class was calculated. Under the

assumption of independence between guesses and that each song was guessed to be steganography correctly with probability $\pi$ the total number $X$ of correct guesses for one respondent could be subjected to a $\chi^2$-test of whether $X$ is binomially distributed with parameters $n = 10$ and $\pi = 0.5$ or not. After merging classes so that the expected number of observations in each class exceeded 2, there were 5 classes and the test statistic turned out 5.8252 rendering a high *p*-value of 0.8171.

So, what does it mean that the null hypothesis $X \in Bin(10, 0.5)$ can not be rejected? Certainly, it does not prove that $X \in Bin(10, 0.5)$ and that $\pi = 0.5$ which corresponds to that respondents can not tell the stego MIDI file apart from the carrier file, only that no deviance in the distribution of $X$ from $Bin(10, 0.5)$ can be found. How large would that deviance have had to be for the hypothesis test to prove it? That question is answered by looking at the power of the test as illustrated in Figure 4. From these curves, it is clear that for deviances of $\pi$ about 0.08 from the null hypothesis value 0.5 the power is clearly greater than 0.95 meaning that the test most likely would have shown a significant difference in this case. For telling even smaller deviances from 0.5 with that great power a larger sample size is needed.

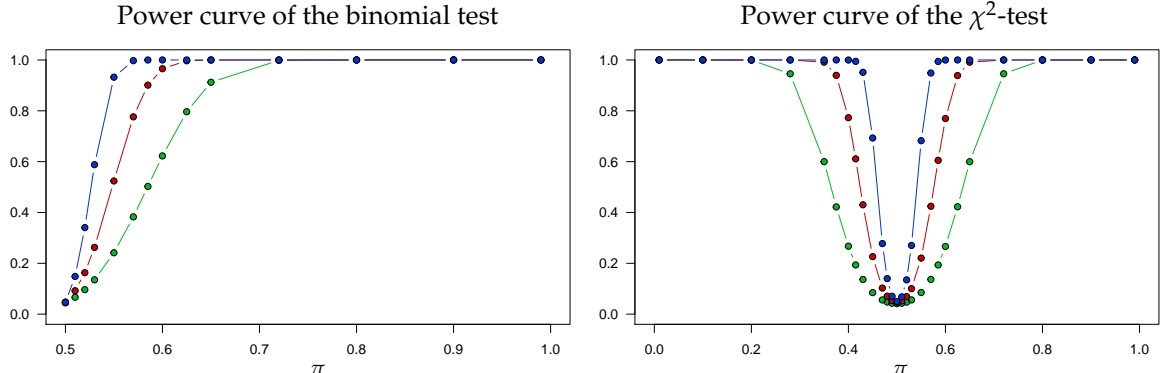

**Figure 4.** To the left: Power curve of the binomial test of deviance from the value 0.5 of the parameter $\pi = P$ (a respondent cannot tell apart the stego MIDI file from the carrier MIDI file.). To the right: Power curve of the $\chi^2$-test of deviance from the binomial distribution with parameters $n = 10$ (since there are 10 questions in the experiment) and $\pi$. For both tests, the power depends on the sample size, i.e., the number of respondents in this case. Here, the sample size was 30 as indicated by the red curves, but had it been 10 the curves would have been as indicated in green and had it been 100 the curves would have been as indicated in blue.

### 3.3. Capacity

For evaluation of steganography methods, Liu and Wu [7] define several variables related to capacity, i.e., the number of bits that can be encoded into the carrier MIDI file referred to as embedding capacity $N_c$, the total number of embedded bits divided by the size of the carrier MIDI file before encoding referred to as the embedding rate $E_r$, the total number of embedded bits divided by the total number of events in the carrier MIDI file referred to as the number of bits/event $N_e$, and the absolute change of the carrier MIDI file before and after encoding divided by the size before encoding referred to as the file-size change rate $F_t$. Following their initiative, the suggested method is evaluated according to these key performance indicators and parameters and compared to the corresponding values of other methods. It is assumed that the full embedding capacity is used for encoding. See Table 3 for a comparison of a selection of steganography methods. As Table 3 shows, the Velody 2 strategy does not limit the number of available events until the $N_e$ approaches 6 bits/event, and even then the reduction is small.

**Table 3.** Table of averages of capacity properties: the number of embedded secret bits $N_b$, number of bits per event $N_e$ and embedding rate $E_r$ as defined by Liu and Wu [7] and briefly explained in the text. Values within parentheses are standard errors.

| Method | Notes | Available | $N_b$ | $N_e$ | $E_r$ |
|---|---|---|---|---|---|
| Velody 2, 4 bits | 2267 (474) | 2267 (474) | 9067 (1896) | 4.00 (0.00) | 6.43% (0.30%) |
| Velody 2, 5 bits | 2267 (474) | 2267 (474) | 11,333 (2370) | 5.00 (0.00) | 8.03% (0.37%) |
| Velody 2, 6 bits | 2267 (474) | 2227 (467) | 13,364 (2804) | 5.91 (0.05) | 9.48% (0.40%) |
| Wu, Hsiang and Chen [8] | 4536 (1999) | 753 (371) | 919 (387) | 0.55 (0.12) | 1.11% (0.22%) |
| Liu and Wu [7] | unknown | unknown | 492 | 1.95 (0.01) | 7.29% [1] |
| Inoue, Suzuki and Matsumoto [3] | unknown | unknown | 3120 (274) | unknown | 4.01% [2] |

[1] According to Liu and Wu [7]. [2] According to Inoue, Suzuki and Matsumoto [3].

It can also be seen that the $E_r$ is high compared to other techniques. While the Velody 2 strategy most likely does not outperform the work of Wu, Hsiang and Chen [8] when it comes to the deviation in average velocity, this is of little practical effect since it according to the statistical experiment seems hard to detect this when listening and that there exist tools that creates exactly this type of deviation as a part of the music production process. The suggested strategy from Liu and Wu [7] achieves a fairly good $N_e$, but this comes at the cost of a low $E_r$ value due to the file size expansion introduced by the strategy. The method proposed by Inoue, Suzuki and Matsumoto [3] achieves a good $E_r$ value with no inflation, performance effects, or obvious artefacts. However, the optimal performance of this elegant strategy is still outperformed by the averaged performance of the Velody 2 method.

**4. Discussion and Conclusions**

A novel MIDI steganography method, called Velody 2, is presented. Its capacity turns out to be on par with the highest capacity methods available in the literature while still leaving few traces of manipulation, such as small values of Mean Absolute Error (MAE) and Peak Signal-to-Noise Ratio (PSNR). It also has no inflation and an experiment was carried out verifying that there are no signs of audible traces.

Regarding many methods suggested in the literature, the MIDI code shows artificial patterns that would not be likely, or even possible, to produce by generating the MIDI file merely by automatic sequencing from sheet music or input of a MIDI song by playing on a keyboard and possibly modifying it slightly afterwards. Examples of such artefacts are extra data not occurring normally in a MIDI song (as is the case with the padding in Liu and Wu [7]), simultaneous note events (so-called simulnotes) that may occur in any order without sounding different and neither causing inflation but where the sequencer always put these events in a certain order and deviance from that pattern should arouse suspicion (in Inoue, Suzuki and Matsumoto [3]), and only two values of velocities (as in Vaske, Weckstén and Järpe [15]). Such artificial giveaways are perfect signals to a steganalyst searching for indications of suspect MIDI steganography.

In the suggested method, velocities are scattered randomly within a narrow interval to be specified. This would have been the result of having played a piece on a keyboard or humanized using a midi software that supports randomization of the velocities. Of course, the mean level is constant and not drifting as is the case in a majority of MIDI music. Still, for harpsichord and organ music this poses no problem at all, and even if this is a strong restriction MIDI music is abundant within this subgroup. An audibility hypothesis test was carried out to see if there were audible traces in the stego MIDI files compared to the carrier files. However, the $p$-values here were 0.3431 (binomial test) and 0.8171 (chi-square test) meaning that no signs of steganography could be detected. If the music is entered by playing the piece on a keyboard this is also likely to cause starting time and duration of notes to be slightly fluctuating. Thus, if the velocities are scattered while starting times and duration are not this might be considered as an unrealistic artefact of the method. However, this is not at all unrealistic taking into account that the composer could well have

quantized the notes after having made the keyboard recording, a very common kind of facility in many MIDI sequencer programs. Thus, the notes would be perfectly according to the measures and bars but velocity differences would remain.

Another aspect of footprint is the file-size change-rate ($F_r$). For Velody 2 this is 0, i.e., there is no change of the file-size at all. This makes it optimal in this respect together with the methods of Inoue, Suzuki and Matsumoto [3] and slightly better than Wu, Hsian and Chen [8]. In addition statistical estimators such as the Mean Absolute Error (MAE) and the Peak Signal-to-Noise Ratio (PSNR) of the various kinds of MIDI events in the music may be interesting from a steganalysis perspective. When having to check a vast material of MIDI music for suspect features, a steganalyst is unlikely to be able to go through each MIDI song's event list to check for revealing footprints such as those mentioned above, unless it is possible to fully automate. Instead, the search is likely to build on summarizing characteristics such as the MAE and PSNR of different MIDI events, and these could systematically be retrieved in an automatized process. Thus, steganography methods which stand out in such a listing are likely to be scrutinized for further indications of steganography. In the proposed method, averages of MAE, ranging from 6.27 to 25.37, and PSNR, ranging from 12.62 to 24.64, were calculated. Corresponding values for other methods have to be calculated to compare methods. This, however, remains as a task for future studies.

It could be argued that most likely a steganography method that creates output that has the reversibility property, thus where the carrier MIDI file can be restored from the stego MIDI file, is by definition not providing plausible deniability. The reason for this is that since the carrier MIDI file can be recreated from the stego MIDI file there has to be extra information available in the file that can be removed.

Veoldy 2 was not developed with robustness in mind. Therefore it does not include any steps to increase its properties in the aspects of robustness. Such development remains as a possibility for future studies.

To further investigate resilience to steganalysis the steganography methods could be submitted to the most common steganalysis tools. This has been done for audio steganography [17] and possibly other kinds of steganography [18] and investigating how successful the procedures are in these papers is an important step to properly finding out the ability of steganography methods to withstand the attempts made by steganalysts. Suggestions for improvements for future experiments include increasing the number of respondents as well as increasing the share of respondents that have training in playing and listening to music. The experiment itself could be improved by generating a large pool of carrier MIDI files and related stego MIDI files from which each experiment randomly generates a unique set of questions. This would reduce the opportunity of collusion leading to test bias.

**Author Contributions:** Formal analysis, M.W.; Investigation, E.J. and M.W.; Methodology, E.J.; Software, E.J. and M.W. All authors have read and agreed to the published version of the manuscript.

**Funding:** The research leading to the results reported in this work received funding from the Knowledge Foundation in the framework of SafeSmart Safety of Connected Intelligent Vehicles in Smart Cities Synergy project (2019–2023), grant number F2019/151.

**Acknowledgments:** The authors which to extend their sincere gratitude to all respondents in the experiment which provided data for the audibility test, see their name in the Appendix A below.

**Conflicts of Interest:** The authors declare no conflict of interest.

### Appendix A

Data and source code is available at https://github.com/wecksten/Velody-2.

The respondents in the audibility experiment were: E. Spennare, T. Holtzberg, P. Wärnestål, M. Dougherty, A. Galozy, A. Olsson, M.R. Bouguelia, J. Johansson, O. Andersson, M.A. Rasool, O. Engelbrektsson, F. Johansson, S. Nilsson, M. Blom, T. Svane, J. Elmlund, A.

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
