# Peer review of "Velody 2—Resilient High-Capacity MIDI Steganography for Organ and Harpsichord Music"

_applsci, doi:10.3390/app11010039_

Round 1

Reviewer 1 Report

This paper investigated how a novel MIDI steganography algorithm can be implemented by manipulation of the velocity attribute subject to restrictions of capacity and security. By restricting the carrier MIDI files to classical organ and harpsichord pieces, the problem of velocities following the mood of the music can be avoided. It turns out that the proposed method, called Velody 2.0, is found to be on par with or better than the cutting edge alternative methods while still possessing a better resilience to steganalysis. An audibility test was conducted to check that there are no signs of audible traces in the stego MIDI files.

Author Response

Thank you for these comments!

Best regards,
Eric Järpe

Reviewer 2 Report

1.Many steganography needs to evaluate robustness (not steganalysis) while it is not proposed in this manuscript. I suggest the authors might propose at least one evaluation in this research.

2.I don't understand the meaning of this sentence "Surprisingly, there is to the author’s knowledge no publically available method for MIDI steganography", does it mean no research papers about MIDI-based steganography? If it does, I think it is a conflict to the next sentence "Many of todays MIDI steganography methods fail to be resilient to steganalysis". If it does not, please explain it.

3.The authors chose organ and harpsichord to deploy their steganography, but why these two types of MIDI? Why not piano or violin or other instruments? Is the method especially good in them? Or something other reasons?

4.Does this method suite to mixed MIDI? Or just applicable on single type of instrument?

Author Response

Dear reviewer,

The authors would like to thank you for your comments and constructive critics.

We have one question:
- The reviewer asks for some aspects of the robustness of the method. However, in none the journal and conference papers in the field of MIDI steganography we found anything about an investigation of the robustness of any method. So, despite this being an important matter -- no question about that -- how should this be assessed? We did find a whitepaper "D.WVL.10 Audio Benchmarking and steganalysis tool", a report within a project ECRYPT European Network of Excellence in Cryptology" from 2006 where the robustness is briefly mentioned but this is about it. Do you have any ideas or recommendation about this? Do you know of any other papers which are dealing more thoroughly with robustness?

Best regards,
Eric Järpe

Reviewer 3 Report

The paper presents a method to embed a crypto message into a MIDI file.

There are a number of major issues in the paper that need to addressed by the authors.

  • It is not clear why the encrypted message is split into blocks: authors should provide an explanation and justify the need for such preparation, that introduces overhead .
  • The experimental section is not convincing and should be improved. First of all, beside subjective evaluation, the authors should also present an objective evaluation of the method, by computing distortion measures, such as PSNR and/or MAE between original and stego MIDI files. Moreover, as they pointed out, a larger number of tests should be performed in order to prove their hypothesis (and this should be done). More details about the parameters used in the experiments should be given in order to be able to replicate the experiments. For instance, length of inserted message, valur of Ne used, etc.
  • In order to prove the steganalysis resilience, the authors should run some steganalyser software for MIDI files (i.e., AAST from AMSL Audio Steganalysis Toolset) and report the results.
  • For the data showed in Table 2, a description of which are better should be given. Moreover, some rows reports percentage value while other just absolute/relative (?) values for the same column. Please use the same measure.

Author Response

Dear reviewer,

We would like to thank you for your comments and constructive critics.

However, we have some questions:
- It is argued that we make a subjective audibility test to indicate if there are effects by the steganography method that could reveal manipulation by leaving audible traces in the stego MIDI files, but we make no objective tests, such as SNR values for the carrier file and compare these to the corresponding values of the stego MIDI files. This is a brilliant idea and a valid point, and we thank you for this! Nevertheless, first, we did not find any such numbers in any of the related literature so we would have no values to compare our method score. Second, the deadline to submit our updated manuscript is only a few days from now, and it is quite impossible to start a new experiment in order to find the SNR values and make "more tests" as you also require for our steganography method. Third, we do have an objective way of measuring effects from the proposed steganography method, namely measuring the change in the average values of velocities from carrier to stego MIDI file and change in the sample variance of velocities from carrier to stego MIDI file. This has been suggested and mentioned in the related literature but we did not see it calculated for any of the other methods. (Also, our Table 2 did not fit the column of variance values so we omitted this from the presentation. Of course, we could include it at the expense of some other column, should you prefer that these values are shown instead.)
- Also, the third reviewer requires some kind of steganalysis assessment since we are claiming that the proposed method offers resilience against steganalysis. Again, this is a very good point and we are very grateful for this justified remark. Nevertheless, it would imply that we make a whole new investigation (especially since the steganalysis methods in the related literature is rather described and defined theoretically rather than implemented and offered as fully functioning and ready-to-use code) which would take several weeks to pursue. Therefore, in spite of being a sharp and valid point, it will have to wait until further studies can investigate this properly. Still, we consider our point that steganography methods should to a larger extent be regarded from a steganalysis perspective is also valid and justified, for the continued development in the field.

Best regards,
Eric Järpe

Round 2

Reviewer 2 Report

The authors improved their manuscript well, and I suggest to accept their paper.

Author Response

Dear Reviewer 2,

We would like to thank you very much for your ambitions readings of our manuscript.

However, we were very puzzled in your current report where it says that there is  "Extensive editing of English language and style required". Is this a mistake or are there some critical langue issues? We are very concerned about attending to all comments so please, let us know how to understand this. Thank you again for all your effort!

Best regards,
Eric Järpe

Reviewer 3 Report

The authors improved quite a lot over the previous version and answered all raised issues.

However, in their effort to discuss the method robustness, they should just say that their method is not robust to attacks, because it has not been developed with that goal in mind.

Author Response

Dear Reviewer 3,

We would like to thank you for your ambitious reading and substantial comments to our manuscript! We now have just one final question about your comment.

Regarding claims of robustness, we now changed that formulation in the Section Discussion and conclusion to:
"Veoldy 2 was not developed with robustness in mind. Therefore it does not include any steps to increase its properties in the aspects of robustness. Such development remains as a possibility for future studies"

In Section The proposed method, we reduced the text about thought on robustness to:
"Another aspect of importance is robustness as considered by e.g. Lang et al [13]. Currently, the proposed method, Velody 2, is not implemented with any support for improving the robustness of the method. Including the hidden message with redundancy in the carrier would not contribute to robustness since the MIDI file would not play and there would not even be an eventlist in the case of as much as one bit failing. However, one could just send the stego MIDI file with redundancy, i.e. sending the same stego MIDI file multiple times. Also, the receiver could be assumed to have many alternative addresses so the transmission could still be made to many different addresses. This way of increasing robustness is claimed to lead to minimal suspicion. Still, this is rather recommended behaviour than a part of the Velody 2 method."

Best regards,
Eric Järpe